# Irreversible Electroporation and Nivolumab Combined with Intratumoral Administration of a Toll-Like Receptor Ligand, as a Means of In Vivo Vaccination for Metastatic Pancreatic Ductal Adenocarcinoma (PANFIRE-III). A Phase-I Study Protocol

**DOI:** 10.3390/cancers13153902

**Published:** 2021-08-02

**Authors:** Bart Geboers, Florentine E. F. Timmer, Alette H. Ruarus, Johanna E. E. Pouw, Evelien A. C. Schouten, Joyce Bakker, Robbert S. Puijk, Sanne Nieuwenhuizen, Madelon Dijkstra, M. Petrousjka van den Tol, Jan J. J. de Vries, Daniela E. Oprea-Lager, C. Willemien Menke-van der Houven van Oordt, Hans J. van der Vliet, Johanna W. Wilmink, Hester J. Scheffer, Tanja D. de Gruijl, Martijn R. Meijerink

**Affiliations:** 1Department of Radiology and Nuclear Medicine, Cancer Center Amsterdam, Amsterdam University Medical Centers, de Boelelaan 1117, 1081 HV Amsterdam, The Netherlands; f.timmer1@amsterdamumc.nl (F.E.F.T.); a.ruarus@amsterdamumc.nl (A.H.R.); e.schouten@amsterdamumc.nl (E.A.C.S.); r.puijk@amsterdamumc.nl (R.S.P.); s.nieuwenhuizen1@amsterdamumc.nl (S.N.); m.dijkstra3@amsterdamumc.nl (M.D.); j.devries1@amsterdamumc.nl (J.J.J.d.V.); d.oprea-lager@amsterdamumc.nl (D.E.O.-L.); hj.scheffer@amsterdamumc.nl (H.J.S.); mr.meijerink@amsterdamumc.nl (M.R.M.); 2Department of Medical Oncology, Cancer Center Amsterdam, Amsterdam University Medical Centers, de Boelelaan 1117, 1081 HV Amsterdam, The Netherlands; j.e.e.pouw@amsterdamumc.nl (J.E.E.P.); j.bakker6@amsterdamumc.nl (J.B.); c.menke@amsterdamumc.nl (C.W.M.-v.d.H.v.O.); jj.vandervliet@amsterdamumc.nl (H.J.v.d.V.); j.w.wilmink@amsterdamumc.nl (J.W.W.); td.degruijl@amsterdamumc.nl (T.D.d.G.); 3Department of Surgery, Amsterdam University Medical Centers, de Boelelaan 1117, 1081 HV Amsterdam, The Netherlands; mp.vandentol@amsterdamumc.nl; 4Lava Therapeutics, Yalelaan 60, 3584 CM Utrecht, The Netherlands

**Keywords:** irreversible electroporation (IRE), nivolumab, IMO-2125, CpG-ODN, metastatic pancreatic cancer, PDAC, checkpoint inhibition, toll-like receptor ligand, PET-C, ablation, immunotherapy

## Abstract

**Simple Summary:**

Metastatic pancreatic ductal adenocarcinoma has a dismal prognosis, and to date no curative treatment options exist. The image guided tumor ablation technique irreversible electroporation (IRE) employs high-voltage electrical pulses through the application of several needle electrodes in and around the tumor in order to induce cell death. IRE ablation of the primary tumor has the ability to reduce pancreatic tumor induced immune suppression while allowing the expansion of tumor specific effector T cells, hereby possibly shifting the pancreatic tumor microenvironment into a more immune permissive state. The addition of immune enhancing therapies to IRE might work synergistically and could potentially induce a clinically significant treatment effect. This study protocol describes the rationale and design of the PANFIRE-III trial that aims to assess the safety of the combination of IRE with IMO-2125 (toll-like receptor 9 ligand) and/or nivolumab in patients with metastatic pancreatic ductal adenocarcinoma.

**Abstract:**

Irreversible electroporation (IRE) is a novel image-guided tumor ablation technique with the ability to generate a window for the establishment of systemic antitumor immunity. IRE transiently alters the tumor’s immunosuppressive microenvironment while simultaneously generating antigen release, thereby instigating an adaptive immune response. Combining IRE with immunotherapeutic drugs, i.e., electroimmunotherapy, has synergistic potential and might induce a durable antitumor response. The primary objective of this study is to assess the safety of the combination of IRE with IMO-2125 (a toll-like receptor 9 ligand) and/or nivolumab in patients with metastatic pancreatic ductal adenocarcinoma (mPDAC). In this randomized controlled phase I clinical trial, 18 patients with mPDAC pretreated with chemotherapy will be enrolled in one of three study arms: A (control): nivolumab monotherapy; B: percutaneous IRE of the primary tumor followed by nivolumab; or C: intratumoral injection of IMO-2125 followed by percutaneous IRE of the primary tumor and nivolumab. Assessments include contrast enhanced computed tomography (ceCT), ^18^F-FDG and ^1^^8^F-BMS-986192 (PD-L1) positron emission tomography (PET)-CT, biopsies of the primary tumor and metastases, peripheral blood samples, and quality of life and pain questionnaires. There is no curative treatment option for patients with mPDAC, and palliative chemotherapy regimens only moderately improve survival. Consequently, there is an urgent need for innovative and radically different treatment approaches. Should electroimmunotherapy establish an effective and durable anti-tumor response, it may ultimately improve PDAC’s dismal prognosis.

## 1. Introduction

Pancreatic ductal adenocarcinoma (PDAC) is one of the most aggressive forms of cancer, with a 5-year overall survival (OS) rate of only 8% [1]. About 30% of newly diagnosed patients present with unresectable locally advanced pancreatic cancer (LAPC) and, despite the introduction of new and more effective chemotherapeutic regimens, responses remain temporary and result in a median OS of 12–14 months [2,3,4]. Several local treatment strategies have been developed that aim to improve this dismal prognosis [5]. Among them is irreversible electroporation (IRE), a non-thermal needle guided ablation technique that utilizes high-voltage electrical pulses to permanently destabilize the phospholipid bilayer of the tumor cell membrane, enforcing apoptotic and delayed necrotic cell death [6,7]. IRE destroys all cells within the ablation zone but preserves extracellular collagenous matrix structures and allows for the cellular regeneration of inlaying both the blood and lymphatic vessels, making it an attractive local treatment option for LAPC [8]. Clinical efficacy studies for the treatment of LAPC [9] revealed the technique to be of additive value to chemotherapy by improving the OS from diagnosis ranging from 15.3 to 27.0 months [5,10,11,12,13,14,15,16,17,18,19,20,21].

### 1.1. Metastatic Disease

However, more than 50% of PDAC patients present with metastatic PDAC (mPDAC), leaving only palliative chemotherapy as a treatment option. With the introduction of FOLFIRINOX (5- fluorouracil, leucovorin, irinotecan, oxaliplatin) in 2012, the prognosis of these patients advanced significantly. Within a trial setting, the median overall survival (OS) improved from 6.8 to 11.1 months [22]. Real-world registry data from the Dutch Cancer Registration (NKR-IKNL) reflected this prognostic improvement but also revealed an attenuated reality with a median OS of 6.4 months in mPDAC patients, including patients unfit for FOLFIRINOX that received ≥1 cycle of chemotherapy [23].

### 1.2. Immune Escape

PDAC has a low mutational burden, resulting in a limited amount of neo-antigens, and its microenvironment is recognized as highly immunosuppressive, allowing the cancer to progress freely [24]. Immune suppression caused by the tumor results from various mechanisms that enable immune cell evasion and exclusion from the tumor microenvironment. Immune evasion is established through the downregulation of antigen presenting pathways (such as MHC-I) by tumor cells, the upregulation of inhibitory immune checkpoint proteins, the restriction of dendritic cell (DC) maturation, and increased apoptotic resistance, thus limiting the release of tumor antigens and immunogenic potential [25,26]. Furthermore, the priming of tumor-specific effector T cells is hampered by active immune suppression through suppressive regulatory T cells (Tregs) and myeloid-derived suppressor cells (MDSC) in the tumor microenvironment [27,28,29]. Collectively, these mechanisms can explain PDAC’s limited response to immunotherapy [30].

### 1.3. Anenestic Effects

Over the past few years, the interventional oncology community has expanded its research focus from focal cytoreduction to the wider systemic influences that local tumor therapy can produce [31]. The widely used term “abscopal” refers to a visual biological or anatomical response that is distant from the treated site, and is historically connoted with radiotherapy [32]. Thus, in this study protocol, we use the terms “enestic” (response in targeted lesion) and “anenestic” (response in non-targeted lesion) to indicate the local and systemic effects after (intratumoral) treatment, as proposed by the European Society for Medical Oncology (ESMO) [33].

### 1.4. Pre-Clinical Evidence IRE Induced Immune Modulation

It is proposed that focal ablation can enhance antigen presentation, provoke inflammation, and reduce tumor-induced immune suppression [34]. Here within, ablation may represent a means to turn PDAC from an immunologically “cold” tumor into a “hot”, immuno-permissive one. Although all ablative modalities may theoretically invoke this mechanism, IRE might possess superior immune potentiating abilities in terms of protein release and T cell activation compared to cryo- or heat ablation [35,36]. Additionally, IRE preserves the larger blood vessels, allowing antigen presenting cells to infiltrate the lesion and transport apoptotic antibodies back to the draining lymph nodes, after which tumor antigen specific T cell activation is induced [37,38].

### 1.5. Clinical Evidence IRE Induced Immune Modulation

Clinical immune monitoring studies of IRE in pancreatic cancer revealed that high-voltage electrical pulses are able to transiently alleviate tumor-induced immune suppression by decreasing the frequency of circulating Tregs [39,40]. This allowed for the simultaneous expansion of activated tumor antigen specific effector T cells, as evidenced by the isolated upregulation of programmed death-1 (PD-1) expression and increased or de novo Wilms Tumor-1 specific responses. The magnitude of the latter effects has been correlated with improved OS [39,40].

### 1.6. Pro-Oncogenic Effects

Similar to the anenestic effects, clinical evidence for pro-oncogenic effects following ablation likewise exists and has been linked to a higher rate of local progression and distant tumor spread in different types of cancer [31]. The exact physiological working mechanism of the immunogenic effects of IRE is not yet fully understood. Therefore, potential pro-oncogenic effects should be taken into consideration. However, based on previous clinical trials in LAPC, pro-oncogenic effects from IRE ablation in PANFIRE-III are deemed unlikely [19,41].

### 1.7. Synergy with Immunotherapy

Although the IRE induced immune response is temporary, it may provide a clinical window of opportunity for potentiation with local or systemic immune enhancing drugs. This treatment combination is termed electroimmunotherapy [42]. PD-1 checkpoint inhibition with nivolumab (Bristol-Meyers Squibb, New York, NY, USA) could release the brakes of the IRE induced effector T cell response to synergistically strengthen the efficacy of both therapies and ultimately establish a durable memory T cell response [43]. Furthermore, there is increasing evidence that the endogenous immune status prior to treatment, influences the outcome of chemo-, radio-, and ablative therapies. High rates of tumor-infiltrating lymphocytes (TILs) and type 1 interferon (IFN) response signatures are linked to higher clinical response rates and represent favorable prognostic factors [44,45,46,47,48]. Such optimal immune priming may be achieved by the peri-ablative administration of IMO-2125 (Idera pharmaceuticals, Exton, PA, USA), a CpG (cytosine-phosphate-guanine) type B oligodeoxynucleotide (CpG-B ODNs) functioning as a synthetic toll-like receptor 9 ligand (TLR9-L) that stimulates DCs [49]. Type 1 IFNs released by properly stimulated plasmacytoid DCs activate tumor infiltrating effector T cells and natural killer (NK) cells while recruiting and activating a myeloid DC subset with superior cross-priming abilities, i.e., cDC1 [46,47,50]. These cDC1s can prime a new generation of tumor antigen-specific effector T cells in the draining lymph nodes and may provide the push needed to kick-start a more durable IRE-induced immune response as well as provide improved responsiveness to the immune checkpoint blockade. See Figure 1 for an illustration of electroimmunotherapy as performed in PANFIRE-III.

### 1.8. Hypothesis

We hypothesize that combining IRE, which reduces immune suppression and stimulates a tumor-specific immune response, with PD-1 checkpoint inhibition using nivolumab and preceded by effective DC priming through intra-tumoral injection of IMO-2125 might establish in vivo immunization and durable treatment results in mPDAC patients.

## 2. Materials and Methods

### 2.1. Objectives

The primary objective of this study is to assess the safety of combining IRE + systemic nivolumab ± intratumoral IMO-2125 in mPDAC patients with at least stable disease after pretreatment with FOLFIRINOX.

The secondary objective aims to assess the (biological) efficacy of IRE + systemic nivolumab ± intratumoral IMO-2125 compared to nivolumab monotherapy in terms of local (enestic) or systemic (anenestic) anti-tumor responses, survival, and quality of life.

### 2.2. Design

The PANFIRE III trial is an investigator initiated, prospective, randomized controlled phase I trial performed in the Amsterdam University Medical Centers under the aegis of the national multidisciplinary Dutch Pancreatic Cancer Group (DPCG). The trial is registered at ClinicalTrials.gov under number NCT04612530. A total of 18 patients will be included and will divided over 3 arms. See Figure 2.

Arm A (control arm): intravenous administration of 240 mg nivolumab every 2 weeks for the first 3 doses followed by intravenous administration of 480 mg every 4 weeks until disease progression.Arm B: percutaneous CT-guided (partial) IRE of the primary pancreatic tumor. After 2 weeks, this will be followed by the intravenous administration of 240 mg nivolumab every 2 weeks for 2 doses followed by intravenous administration of 480 mg every 4 weeks until disease progression.Arm C: single intratumoral (i.t.) injection of 8 mg IMO-2125, which will be followed by percutaneous CT-guided (partial) IRE of the primary pancreatic tumor after one week. A 240 mg dose of nivolumab is administered intravenously every 2 weeks for 2 doses, which will begin two weeks after IRE, followed by the intravenous administration of 480 mg every 4 weeks until disease progression.

### 2.3. Eligibility Criteria

To be eligible, adult patients require radiologically and histologically proven mPDAC (AJCC stage IV). A maximum of 5 unequivocal metastases of ≥1 cm at the time of inclusion (i.e., after pretreatment with FOLFIRINOX) are allowed. Patients should have at least stable disease (according to RECIST version 5.0) after a minimum of 8 courses of FOLFIRINOX and must be in good clinical condition (WHO 0–2). A minimum of a 4-week interval between the final cycle of chemotherapy and start of the study-related treatment is required. The in- and exclusion criteria are summarized in Table 1. Prior to inclusion, all patients will be discussed in a multidisciplinary hepatopancreaticobilliary (HPB) tumor board consisting of a hepatogastroenterologist, hepatobiliary surgeon, medical oncologist, radiation oncologist, and abdominal and interventional radiologist. Decision on PANFIRE-III trial participation will be at their discretion. All participants will provide written informed consent.

### 2.4. Interventions

#### 2.4.1. Percutaneous CT-Guided IRE

A total of 12 patients (*n* = 6 in arm B and *n* = 6 in arm C) will receive IRE treatment of the primary tumor under general anesthesia induced with propofol, sufentanil, and rocuronium and maintained with propofol and remifentanil. Antibiotic prophylaxis will be administered within one hour prior to the procedure (Cefuroxim 1500 mg i.v. and Metronidazole 500 mg i.v.). A pre-procedural 4-F straight flush catheter (Cordis Corporation, Bridgewater, NJ, USA) flush catheter will be introduced via the right common femoral artery in the abdominal aorta up to a supraceliac level for per-procedural transcatheter CT arteriography [41]. All ablations will be performed percutaneously using ceCT guidance and will be employed using the NanoKnife system under ECG-gating (AccuSync model 72; AngioDynamics, Latham, New York, USA). A defibrillator will be present at all times. A pre-procedural catheter guided ceCT scan (unenhanced, arterial (7 s), and early portal venous phase (22 s)) will be performed for tumor staging confirmation, exact tumor size measurement, and electrode planning. Depending on tumor size, 2–6 needle electrodes will be placed in the bulky part of the tumor in a parallel fashion while aiming at an inter-electrode distance of 20 mm (±2 mm) with at least a 5 mm margin from adjacent critical structures [53]. The generator-based tumor free margin will be set at 5 mm, and the working length of the electrodes will be set at 15 mm. Ablations may be intentionally incomplete, aiming for antigen release rather than optimal cytoreduction and emphasizing patient safety. The ablation protocol consists of 10 test pulses of 1500 V/cm and 90 μs per electrode pair followed by a sequential pulsing scheme of 90 additional pulses per electrode pair. The voltage per centimeter setting can be adjusted in 10% steps in case of a pending under- or overcurrent (<20 A or >45 A, respectively). Immediately after IRE, a control transcatheter arteriography abdominal CT scan will be made to evaluate possible early complications. Additional periprocedural interventions to enhance safety are allowed such as placement of biliary endoprosthesis, percutaneous transhepatic cholangiography drainage, or arterial/portal venous stenting in case of an impending IRE-induced occlusion.

#### 2.4.2. Anti-PD-1 Monoclonal Antibody (mAb) (Nivolumab)

All patients (*N =* 18) will initially receive 240 mg nivolumab (dissolved in 250 mL NaCl 0.9%) administered intravenously every 2 weeks—3 doses in arm A and 2 doses in arms B and C—followed by 480 mg nivolumab (dissolved in 500 mL NaCl 0.9%) administered intravenously every 4 weeks. Nivolumab will be administered until unequivocal disease progression. The infusion time will be 30–60 min. Blood will be drawn prior to every treatment cycle, verifying hematological and endocrinological values, tumor marker CA19.9, electrolytes, albumin, and kidney and liver function. See Table 2.

#### 2.4.3. CpG Oligodeoxynucleotide (IMO-2125)

All patients in arm C (*n* = 6) will receive a single intratumoral injection with 8 mg of IMO-2125 dissolved in 1 mL NaCl 0.9%. Injection will be administered in a daycare setting under local anesthesia and will be performed percutaneously using CT and/or ultrasound guidance. IMO-2125 solution will be injected with a 21-gauge stainless steel disposable needle with side holes indicated for the infusion and aspiration of fluids (ProFusion™, Cook, Bloomington, IN, USA).

### 2.5. Outcome Measures

The primary objective (safety) will be assessed by recording adverse events (AEs) and serious adverse events (SAEs) directly associated with each treatment arm occurring up to 90 days after any of the interventions according to the Common Terminology Criteria of Adverse Events (CTCAE) v5.0 [54].

The secondary objective ([biological] efficacy) will be assessed using biochemical responses (tumor marker CA19.9), immunological responses (flow cytometry, histopathology/immunohistochemistry (IHC), proteomics), radiological responses (contrast-enhanced computed tomography (ceCT)), nuclear radiological responses (^18^F-fluorodeoxyglucose (FDG), and ^1^^8^F-BMS-986192 (Programmed Death- Ligand 1 (PD-L1)) positron emission- computed tomography (PET) CT imaging) [55]. Furthermore, quality of life and pain questionnaires, overall survival (OS), and progression-free survival (PFS) will be recorded. Inherent to the phase I trial design and small study sample, OS and PFS are specifically evaluated subordinate to and in relation to the previously noted outcome measures.

### 2.6. Data Collection and Analysis

#### 2.6.1. Interim Safety Analysis

To ensure safety and to monitor toxicity of the electroimmunotherapeutic treatment combinations a 3 + 3 step up design will be implemented. The study will be split into 4 phases, separated by interim safety analyses: phase 1 (inclusion 1–6), phase 2 (inclusion 7–12), phase 3 (inclusion 13–15), and phase 4 (inclusion 16–18). Randomized inclusion will start in arms A and B (phases 1 and 2) at the same time to minimize selection bias and to secure optimal qualitative immunological analyses of the added value of IRE to nivolumab in mPDAC. Inclusion in arm C (phases 3 and 4) will start after the finalization of the confirmed safety of arms A and B after the second interim analysis. Interim safety analyses will be assessed by the following stopping rules based on recommendations of the clinical trial design task force of the National Institutes of Health Investigational Drug Steering Committee [56]: if >50% of patients within one study arm per study phase develop a grade 5 drug- or IRE-related SAE (i.e., death) within 30 days after treatment and/or if >50% of all included patients overall develop a grade 4 drug- or IRE-related SAE within 30 days after treatment, the study will be terminated.

#### 2.6.2. Survival

Overall survival (OS) is defined as the date of the first study-related treatment (T = 0) until the date of death from any cause. Progression-free survival (PFS) is defined as the time from the first study-related treatment until the date of unequivocal disease progression according to iRECIST [57] and PERCIST [58] criteria.

#### 2.6.3. Blood and Tissue

Systemic and local immune responses will be assessed by sampling venous blood and primary and metastatic tumor tissue in the same session. Samples will be collected at 3 different time points: baseline, 2 (arm A, B) or 3 (arm C) weeks, and 5 (arm A, B) or 6 (arm C) weeks after the first study-related treatment. In arm C, extra blood samples will be collected 1 week after study start to assess the systemic immune response after initial intratumoral injection with CpG. Blood will be examined by flow cytometry for changes in frequency and the activation/proliferation status of several suppressive and permissive immune-cell subsets (memory/effector CD4^+^ and CD8^+^ T cells, Tregs, B cells, DCs, NK cells and MDSCs). Tissue will be analyzed using flow cytometry (tumor-infiltrating lymphocytes (memory/effector CD4^+^ and CD8^+^ T cells, Tregs) and NK cells), histopathology/IHC (microsatellite instability, tumor-infiltrating lymphocytes, myeloid infiltration (M2 macrophages, DCs) and tumor markers), and proteomic analysis. Biochemical responses will be assessed by the evaluation of tumor marker CA19.9 in venous blood drawn at baseline and prior to every nivolumab treatment cycle. See Figure 3 for the timing of the study assessments.

#### 2.6.4. Imaging

Radiological responses will be assessed by ceCT (SOMATOM Sensation or Drive, Siemens AG, München, Germany), and nuclear radiological responses will be assessed by ^18^F-FDG PET-CT, and ^18^F-BMS-986192 (Bristol Meyer Squibb) PET-CT (Philips Gemini TF PET-CT system, Philips Medical Systems, Cleveland, OH, USA) and performed at 3 different time points: baseline, 2 (arm A, B) or 3 (arm C) weeks, and 5 (arm A, B) or 6 (arm C) weeks after treatment start. A third ceCT and ^18^F-FDG PET-CT will be performed 3 months after the start of treatment in all study arms.

CeCT imaging (abdomen and thorax) is conducted according to EANM guidelines to assess viable tumor lesions, enestic tumor response (primary lesion), and anenestic tumor response (metastatic lesion) (see discussion) [59]. Treatment response to ceCT is assessed using the immune response evaluation criteria in solid tumors (iRECIST) [57,60]. iRECIST utilizes an intermediary progressive state known as unconfirmed progressive disease (iUPD), in which a radiological increase in the sum of the target lesions (>20%; longest diameter; axial plane), unequivocal progression of existing non-target lesions, or the appearance of one or more new lesions are identified. The unconfirmed progressive disease is only confirmed (iCPD) if at the subsequent imaging timepoint, the target sum has further increased (≥5 mm) or if there is additional unequivocal progression of non-target lesions and/or the appearance of new lesions (sum ≥ 5 mm). However, if the subsequent imaging after iUPD reveals stable disease (SD), partial response (PR), or complete response (CR) (compared with baseline), the iUPD status is reset to the newly acquired status.

Changes in ^18^F-FDG and ^18^F-BMS-986192 uptake in tumor lesions and organs of interest (as measured both visually and semi-quantitatively by means of different standardized uptake values (SUVs)) from baseline to early and late treatment will be investigated and correlated with radiological response and blood and tissue analyses. Whole body PET (vertex to mid-thigh) will start 60 min after tracer injection followed by a ceCT for attenuation correction and the anatomical correlation of ^18^F-FDG and ^18^F-BMS-986192 PET. To prevent remaining ^18^F signal at the time the second scan, a minimum delay of 10 half-lives (19 h) will be ensured. In order to validate the usage of the derived image, SUVs of the ^18^F -BMS-986192 tracer venous blood sampling will be performed at 5, 10, 20, 30, 40 and 55 (±5) minutes post injection.

#### 2.6.5. Questionnaires

Pain and quality of life will be assessed using validated visual analog score (VAS) questionnaires and Quality of Life Questionnaires (QLQ) at baseline and every three-months as part of the PACAP-trial (NCT03513705) [61].

### 2.7. Follow-Up

Patient follow-up after the first 3 months will consist of blood sampling prior to every consecutive nivolumab administration (see Table 2), ceCT imaging every 3 months, and quality of life and pain questionnaires every 3 months. Study medication will be stopped after 12 months or when unequivocal disease progression occurs. The trial will end twelve months after the inclusion of the final patient.

### 2.8. Data Collection and Handling

Data will be collected by the study coordinators and will be treated confidentially and anonymously. A subject identification code will be used to link the data to the subject. The study coordinators safeguard the key to the identification code. The handling of personal data complies with the Dutch Personal Data Protection Act. See Figure 4 for overview of all analysis modalities.

### 2.9. Sample Size Calculation and Statistical Considerations

In this phase I trial, a sample size of 6 patients per treatment arm and the 3 + 3 step-up design are based on establishing early safety data. No power analysis was performed as efficacy is not the primary study objective.

## 3. Discussion

### 3.1. Preclinical Evidence of Synergy

While the exact mechanism of action underlying systemic effects of IRE is not fully understood, it most likely involves the release of antigens and the activity of DCs, as they regulate tolerance versus immunity, depending on their maturation state [62,63]. Exposure to sufficient quantities of antigens and maturation signals, i.e., damage associated molecular patterns (DAMPs), leaves DCs capable of upregulating their T cell stimulatory function. Ablation results in massive tumor cell death [37] and can provoke an immune cascade of tumor antigen and DAMP release, DC activation, and tumor antigen uptake followed by migration to the lymph nodes, where effector T cell cross-priming can take place [64,65,66]. Activated T cells may subsequently home to loco-regional and distant sites to eliminate (micro-)metastases producing an anenestic effect [67]. Research in an immunocompetent PDAC murine model that leveraged this ablation-induced immune response by combining IRE with anti-PD-1 mAbs provided encouraging outcomes in terms of immune activation, tumor regression, and improved survival [43]. Furthermore, it was observed that priming the tumor micro-environment with the injection of CpG, 7 days prior to ablation, could further enhance effective synergistic tumor control in an epithelial cancer mouse model [68].

### 3.2. Clinical Evidence of Synergy

The first clinical trials combining IRE with immunotherapy, such as adoptive allogenic natural killer cell therapy and PD-1 checkpoint inhibition, were actively pursued in end stage pancreatic cancer patients and the combinations were proven to be safe with promising preliminary efficacy results [69,70].

### 3.3. Timing of Study Interventions

The specific timing and sequence of IMO-2125, IRE, and Nivolumab administration in PANFIRE-III are based on observations of immune cell kinetics in previous studies. Simultaneous Treg downregulation and tumor specific CD4 and CD8 T cell expansion with increased PD-1 expression peak 14 days after ablation in IRE treated PDAC patients [40]. Hence, the onset of Nivolumab administration 14 days after IRE ablation in arm B and C to potentially release the brakes on this effector T cell response. Earlier discussed preclinical findings on DC priming are in accordance with clinical findings of optimally increased DC frequency and maturation status, in both the circulation and sentinel lymph nodes, 7 days after CpG administration [71,72]. Consequently, IMO-2125 is intratumorally administered 7 days prior to IRE ablation aiming to secure optimal DC priming at the time of ablative antigen release.

The addition of immunotherapeutic agents to FOLFIRINOX is also proposed to strengthen and synergistically increase anti-tumor effects [73]. Systemic chemotherapy itself can promote the release of tumor antigens and may activate anti-cancer immunity to enhance tumor growth suppression [74,75]. In this regard, pre-treatment with FOLFIRINOX in this trial might add to the improved tumor sensitivity to nivolumab in all treatment arms. PANFIRE-III aims to gain new insights into the potentially synergistic pathways of both ablation and chemotherapy in combination with immunotherapeutic agents in mPDAC.

### 3.4. Immune Response

This study’s flow cytometric, histopathologic/IHC, and proteomic analyses of primary and metastatic biopsy samples will provide unique insights into the different mechanisms that drive the local and systemic immune response to achieve the desired enestic and anenestic effects in mPDAC. The flow cytometric analyses of blood samples will allow comparison of the development of systemic innate and adaptive immune responses after each treatment and over time. Assessments of the post treatment immune responses are specifically timed for optimal comparison between the treatment arms. Blood and tissue are sampled 2 (arm A and B) or 3 weeks (arm C) and 5 (arm A and B) or 6 weeks (arm C) after start of the treatment for the assessment of the innate immune response to each individual treatment (arm A: nivolumab, arm B: IRE, arm C: CpG + IRE) and the adaptive immune response to each individual treatment in combination with Nivolumab, respectively. To optimally compare the immune effects induced by IRE to the addition of CpG and/or Nivolumab, identical timing between IRE ablation, blood and tissue sampling, and imaging in arm B and C were given precedence over the uniform timing of data collection. Hence, sampling in arm C is deferred by 1 week. See Figure 3 for the timing of blood and tissue sampling.

If electroimmunotherapy proves safe, potential radiological, immunological, or biochemical responses generated in this study will provide new information that will be valuable for the optimization of combination treatment strategies for mPDAC.

### 3.5. Imaging

The PANFIRE-III trial primarily focuses on the typical ceCT and ^18^F-FDG PET-CT imaging characteristics of tumor tissue over time and during follow-up. Although the performance of ^18^F-FDG PET in the follow-up of PDAC after treatment with immunotherapeutic drugs is unexplored, a meta-analysis of the use of ^18^F-FDG PET in combination with ceCT for detection of recurrent disease reported a reasonable diagnostic accuracy with pooled estimates for sensitivity and specificity of 95% and 81%, respectively [76]. Additionally, preclinical evidence supports the potential of ^18^F-FDG PET to monitor checkpoint inhibitors associated metabolic changes in lymphoid organs [77].

However, there is a need to improve PDAC detection, prognosis prediction, and treatment response evaluation. Conventional imaging techniques such as ceCT, MRI or EUS do not always offer a reliable differentiation between PDAC and benign conditions or between viable tumor and necrotic tissue after neo-adjuvant chemotherapy, nor do they provide insight into the immune status of the tumor microenvironment and the lymphoid organs. Additionally, the emerging prescription of immunotherapies warrants new and accurate diagnostic techniques for response evaluation. PET imaging with novel tracers such as the PD-L1 targeting ^18^F-BMS-986192 tracer [78] could offer an innovative approach for these hurdles. Imaging with ^18^F-BMS-986192 was previously studied in non-small cell lung cancer patients prior to treatment with anti-PD-1 mAbs and was proven to be feasible and safe [55]. It was found that patients responding to treatment with anti-PD-1 mAbs experienced higher ^18^F-BMS-986192 uptake than non-responders. No clinical PET studies with PD-L1 tracers have been performed in pancreatic cancer patients yet. Hence, the incorporated explorative ^18^F-BMS-986192 tracer study is primarily used for tracer validation in metastatic pancreatic cancer patients, as it will generate qualitative and quantitative information about the tracer’s uptake in the tumor and lymphoid organs. Additionally, it might provide new insights in innate and adaptive immune dynamics following each treatment combination. Indeed, type-1 IFN release by pDC upon CpG binding may induce PD-L1 expression in both tumor and immune cells; additionally, T cells that are recruited to and are activated in the tumor microenvironment may induce PD-L1 expression through IFNγ release. It should be noted that the amount of delivered ^18^F-BMS-986192 is in the nanomolar quantity and is far below the dose required for pharmacological effects.

## 4. Conclusions

Electroimmunotherapy through a combination of IRE with local and systemic immunotherapeutic agents may provide efficient in vivo immunization against pancreatic cancer. PANFIRE-III aims to assess the safety and (biological) efficacy of combination therapy with IRE, nivolumab, and IMO-2125 in patients with mPDAC that have been effectively pre-treated with FOLFIRINOX.

The study treatment might allow the immune system to initiate systemic tumor degradation and protection against further tumor growth and spread. When proven safe and clinically reproducible, future research should be pursued to optimize the dosage and timing of drug administration, IRE pulse delivery settings, and choice of immunotherapeutic agents. If electroimmunotherapy can truly be used to trigger a systemic anti-tumor effect and can incite a durable response in patients with mPDAC, IRE may provide the missing link between local and systemic treatment in pancreatic cancer.

## Figures and Tables

**Figure 1 cancers-13-03902-f001:**
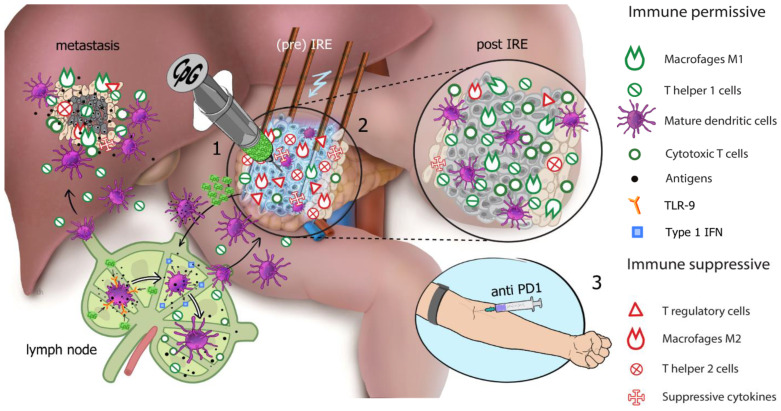
Electroimmunotherapy. Illustrated is the proposed working mechanism of electroimmunotherapy and the three separate treatment stages to achieve synergistic immune modulation as performed in PANFIRE-III study arm C. The primary pancreatic tumor maintains an immunosuppressive microenvironment with the presence of regulatory T cells (Tregs), M2 Macrophages, T helper 2 cells, and suppressive cytokines (red icons), while the effector T cells (T helper 1 cells and cytotoxic T cells) are downregulated. Stage (1): Priming of the tumor microenvironment. CpG (cytosine-phosphate-guanine) type B oligodeoxynucleotide (CpG-B ODN) is injected into the primary tumor and diffuses to the primary tumor-draining lymph nodes where it binds to toll-like receptor 9 (orange receptor) on plasmacytoid dendritic cells (DCs), which mature and release type-1 IFN, which in turn can activate lymph node-resident conventional DCs (cDCs), causing these to mature (purple cells). Effective DC maturation results in their improved ability for tumor antigen uptake and presentation and stimulates type-1 IFN release (blue icons), resulting in the activation of cytotoxic- and helper T cells (green icons). Stage (2): Ablative antigen release and downregulation of tumor induced immune suppression. Irreversible electroporation (IRE) of the primary pancreatic tumor causes massive cell death resulting in antigen release (black dots). Antigens are taken up by DCs and transported back to the lymph nodes for T cell cross priming to result in adaptive tumor specific T cell responses (green icons). As IRE reduces the tumor mass, it reduces the secretion of immunosuppressive cytokines and consequently reduces numbers of circulating suppressive immune cell subsets (red icons). The tumor microenvironment shifts from immune suppressive (pre-IRE, red icons) to immune permissive (post-IRE, green icons). Stage (3): Enhancing the induced effector T cell response by intra venous injection of the anti-PD-1 monoclonal antibody (mAb). PD-L1 on the cancer cell surface binds to PD-1 on the T cell surface, inhibiting immune cell activity. Anti-PD-1 mAb (PD-1 checkpoint inhibitor) binds to the PD-1 receptors on the T cells, thereby blocking the receipt of inhibitory signals via PD-L1 and allowing the T cells to uninhibitedly attack the tumor cells (not illustrated). NB: new insights suggest that the PD-1 blockade can also release co-stimulatory signaling in T cells through CD28, leading to increased priming in tumor-draining lymph nodes. Thus, the combination of all three treatment modalities (IRE, CpG, Anti-PD-1) may work synergistically and together may alter the tumor microenvironment to induce a systemic immune response and ultimately cause an “anenestic” effect in distant metastatic lesions (illustrated in the liver). (Reprinted with permission of Geboers et al. [51].)

**Figure 2 cancers-13-03902-f002:**
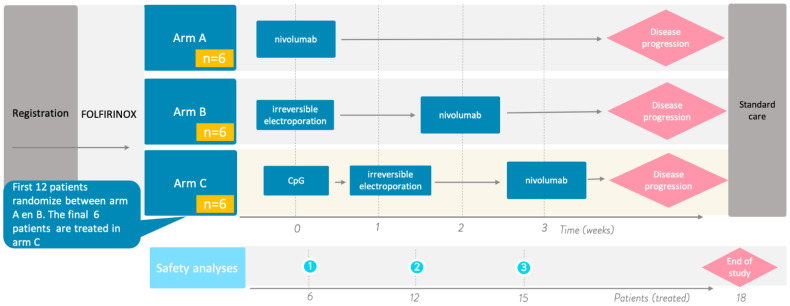
PANFIRE III treatment schedule. All patients require pretreatment with a minimum of 8 cycles of FOLFIRINOX, with at least stable disease, prior to PANFIRE III inclusion. The first 12 patients will be randomized to treatment arm A or B. Arm C will open for the last 6 patients upon the validation of the interim safety analyses. These safety analyses will be performed after the treatment of 6, 12, and 15 patients. Arm A: nivolumab monotherapy. Arm B: irreversible electroporation succeeded by intravenous administration of nivolumab. Arm C: priming with intratumoral injection of CpG (IMO-2125), followed by irreversible electroporation after 1 week. This is succeeded by intravenous administration of nivolumab two weeks later. In all treatment arms, nivolumab will be administered until unequivocal disease progression.

**Figure 3 cancers-13-03902-f003:**
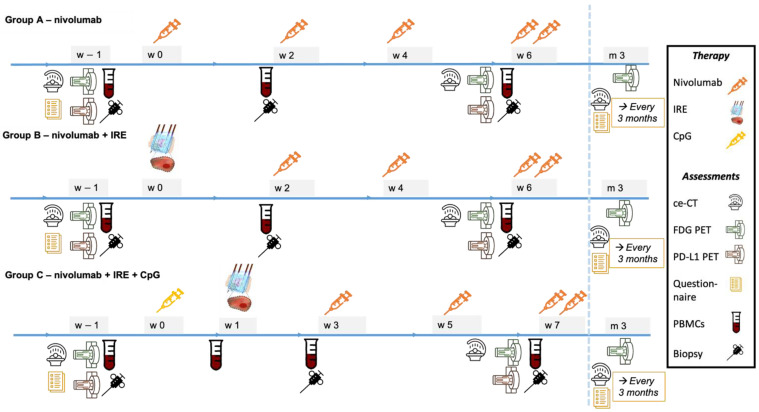
PANFIRE III specified treatment schedule and study assessments per study arm. Arm A: nivolumab monotherapy; intravenous administration of 240 mg nivolumab every 2 weeks for 3 doses followed by intravenous administration of 480 mg nivolumab every 4 weeks. Assessments: ceCT (baseline, 5 weeks, and after every 3 months), FDG and PD-L1-PET (baseline and 5 weeks), blood and biopsy sampling (baseline, 2 weeks (prior to second nivolumab administration) and 5 weeks (prior to fourth nivolumab administration)), and QOL questionnaires (baseline and every 3 months). Arm B: irreversible electroporation succeeded by intravenous administration of 240 mg nivolumab every 2 weeks for 2 doses followed by intravenous administration of 480 mg nivolumab every 4 weeks. Assessments: ceCT (baseline, 5 weeks, and after every 3 months), FDG and PD-L1-PET (baseline and 6 weeks), blood and biopsy sampling (baseline, 2 weeks (prior to second nivolumab administration) and 5 weeks (prior to fourth nivolumab administration)), and QOL questionnaires (baseline and every 3 months). Arm C: priming with intratumoral injection of IMO-2125 after 1 week followed by irreversible electroporation. This is succeeded 2 weeks later by intravenous administration of 240 mg nivolumab every 2 weeks for 2 doses followed by intravenous administration of 480 mg nivolumab every 4 weeks. Assessments: ceCT (baseline, 6 weeks, and after every 3 months), FDG and PD-L1-PET (baseline and 6 weeks), blood and biopsy sampling (baseline, 1 week, 3 weeks (prior to first nivolumab administration), and 6 weeks (prior to third nivolumab administration)), and QOL questionnaires (baseline and every 3 months). In all treatment arms, nivolumab will be administered until unequivocal disease progression. W: week, M: month, IRE: irreversible electroporation, ceCT: contrast enhanced computed tomography, PET: positron emission tomography, FDG: fluorodeoxyglucose, PD-L1 PET: programmed death—ligand 1. Double syringe nivolumab at 6 (arm A/B) and 7 (arm C) weeks illustrates change of nivolumab dosing from 240 mg fortnightly to 480 mg monthly.

**Figure 4 cancers-13-03902-f004:**
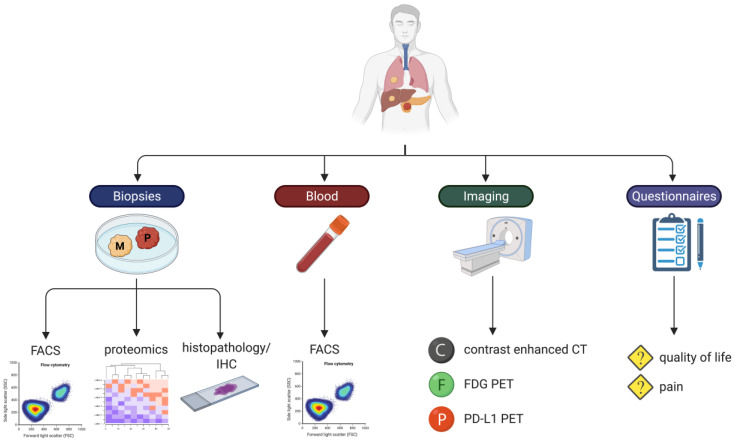
Data collection and analyses per patient in PANFIRE-III. Biopsies from primary (P) and metastatic (M) tissue will be analyzed using flow cytometry, histopathology/immunohistochemistry (IHC), and proteomics. Blood samples will be analyzed using flow cytometry. Imaging will be performed by ceCT, FDG (^18^F-FDG) PET-CT, and PD-L1 (^18^F-BMS-986192) PET-CT. Pain scores and quality of life scores will be evaluated by validated questionnaires.

**Table 1 cancers-13-03902-t001:** In- and exclusion criteria PANFIRE-III.

Inclusion	Exclusion
Radiologically and histopathologically proven stage IV pancreatic cancer (according to the AJCC staging system for pancreatic cancer [52]).	Brain metastases.
Max. 5 unequivocal metastases ≥ 1 cm at the time of inclusion (i.e., after FOLFIRINOX).	Active epilepsy (last convulsion < 5 years).
Primary tumor is in situ.	History of cardiac disease: – Congestive heart failure > NYHA Class 2. – Active coronary artery disease (defined as myocardial infarction within 6 months prior to screening). – Ventricular cardiac arrhythmias requiring anti-arrhythmic therapy or pacemaker (beta-blockers for antihypertensive regimen are permitted; atrial fibrillation is not contra-indicated).
A minimum of 8 cycles of FOLFIRINOX chemotherapy is required before study inclusion, with at least stable disease according to RECIST.	Known hypersensitivity to any oligodeoxynucleotides.
Age ≥ 18 years.	Compromised liver function defined as warning signs of portal hypertension, INR > 1,5 without use of anticoagulants, bilirubin > × 1.5 Upper limit of normal range (ULN) ASAT > 3.0 × ULN, ALAT > 3.0 × ULN.
World Health Organization (WHO) scale performance status 0–2.	Compromised kidney function defined as eGFR < 30 mL/min (using the Cockcroft Gault formula).
Adequate bile drainage in case of biliary obstruction.	Active autoimmune disease requiring disease-modifying therapy at the time of screening, i.e., >10 mg prednisolone per day or equivalent to this regimen.
	Uncontrolled hypertension. Blood pressure must be ≤160/95 mmHg at the time of screening on a stable antihypertensive regimen.
	Uncontrolled infections (>grade 2 NCI-CTC version 3.0) requiring antibiotics.
	Immunotherapy prior to the procedure for the treatment of cancer.
	Previous surgical therapy for pancreatic cancer.
	Second primary malignancy with median 5-year OS < 90%. This excludes adequately treated cancers such as non-melanoma skin cancer, in situ carcinoma of the cervix uteri, superficial bladder cancer, or other malignancies that have been previously treated without signs of recurrence.
	Allergy to contrast agent.
	Allergy to PET tracers 18F-FDG and 18F-BMS-986192.
	Any implanted stimulation device.
	Portal vein or VMS stenosis > 70%, or any arterial stenosis (superior mesenteric artery, celiac artery, common hepatic artery) > 70% unless effectively stented.
	Any condition that is unstable or that could jeopardize the safety of the subject and their compliance in the study.

**Table 2 cancers-13-03902-t002:** Required blood tests prior to Nivolumab administration PANFIRE-III.

Lab Test	Prior to First Cycle of Nivolumab	Prior to Consecutive Cycles of Nivolumab
**Full blood:**	hemoglobin/leukocytes and differentiation/thrombocytes	hemoglobin/leukocytes and differentiation/thrombocytes
**Electrolytes:**	natrium/potassium/calcium/magnesium/phosphate	natrium/potassium
**Liver function:**	albumin/glucose/lipase/bilirubin/Alkaline phosphatase/γ-glutamine transferase/aspartate-aminotransferase/alanine-aminotransferase/lactate-dehydrogenase	albumin/glucose/lipase/bilirubin/Alkaline phosphatase/γ-glutamine transferase/aspartate-aminotransferase/alanine-aminotransferase/lactate-dehydrogenase
**Kidney function:**	creatinine/urea	creatinine
**Thyroid function:**	thyroid stimulating hormone/thyroxin	thyroid stimulating hormone/thyroxin
**Acute phase proteins:**	c-reactive protein	c-reactive protein
**Hormones:**	cortisol/luteinizing hormone/follicle stimulating hormone/adrenocorticotropic hormone	
**Tumor markers:**	CA19.9	

## Data Availability

Not applicable.

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
