# Peer review of "Irreversible Electroporation and Nivolumab Combined with Intratumoral Administration of a Toll-Like Receptor Ligand, as a Means of In Vivo Vaccination for Metastatic Pancreatic Ductal Adenocarcinoma (PANFIRE-III). A Phase-I Study Protocol"

_cancers, 2021, doi:10.3390/cancers13153902_

Round 1
Reviewer 1 Report
This is a study design / protocol rather than original research. I feel the methodology is sound. Please expand Fig. 1 legend.
Author Response
Point 1: This is a study design / protocol rather than original research. I feel the methodology is sound. Please expand Fig. 1 legend.
Response 1: We want to thank reviewer 1 for his/her review and recommendation. We have expanded the legend belonging to figure 1 in order to describe the proposed synergistic working mechanism of the treatment combinations in more detail from line 169 to line 228:
Illustrated is the proposed working mechanism of electroimmunotherapy and the three separate treatment stages to achieve synergistic immune modulation as performed in PANFIRE-III study arm C. The primary pancreatic tumor maintains an immunosuppressive microenvironment by the presence of regulatory T cells (Tregs), M2 Macrophages, T helper 2 cells and suppressive cytokines (red icons) while effector T cells (T helper 1 cells and cytotoxic T cells) are downregulated. Stage (1): Priming of the tumor microenvironment. CpG (cytosine-phosphate-guanine) type-B oligodeoxynucleotide (CpG-B ODN) is injected in the primary tumor and diffuses to the primary tumor-draining lymph nodes where it binds to toll-like receptor 9 ligand (orange receptor) on plasmacytoid dendritic cells (DCs), which mature and release type-1 IFN, which in turn can activate lymph node-resident conventional DCs (cDCs) causing these to mature (purple cells). Effective cDC maturation results in their improved ability for tumor antigen uptake and presentation, resulting in activation of cytotoxic- and helper T cells (green icons). Stage (2): Ablative antigen release and downregulation of tumor induced immune suppression. Irreversible electroporation (IRE) of the primary pancreatic tumor causes massive cell death resulting in antigen release (black dots). Antigens are taken up by DCs and transported back to the lymph nodes for T cell cross priming to result in adaptive tumor specific T cell responses (green icons). As IRE reduces the tumor mass it reduces the secretion of immunosuppressive cytokines and consequently reduces numbers of circulating suppressive immune cell subsets (red icons). The tumor microenvironment shifts from immune suppressive (pre-IRE, red icons), to immune permissive (post-IRE, green icons). Stage (3): Enhancing the induced effector T cell response by intravenous injection of anti-PD-1 monoclonal antibody (mAb). PD-L1 on the cancer cell surface binds to PD-1 on the T cell surface, inhibiting immune cell activity. Anti-PD-1 mAb (PD-1 checkpoint inhibitor) binds to the PD-1 receptors on T cells thereby blocking the receipt of inhibitory signals via PD-L1 and allowing the T cells to uninhibitedly attack tumor cells (not illustrated). NB: new insights suggest PD-1 blockade can also release co-stimulatory signaling in T cells through CD28, leading to increased priming in tumor-draining lymph nodes. Thus, the combination of all 3 treatment modalities (IRE, CpG, Anti-PD-1) may work synergistically and together alter the tumor microenvironment to induce a systemic immune response and ultimately cause an “anenestic” effect in distant metastatic lesions (illustrated in the liver).

Reviewer 2 Report
Geboers and al. present an original Phase I trial, randomized and controlled devoted to pancreatic cancer patients with a stage IV tumor and at least as a second phase treatment. 3 arms have been designed A: nivolumab as control; B: nivolumab + tumor electroporation; C: nivolumab + tumor electroporation after IV injection of TLR ligand (PANFIRE-III – clinical trial # NCT04612530). The primary end point is safety, secondary are: OS, PFS, immuno-monitoring data on “biopsies, blood and imaging”, QoL.
The protocol is clearly presented with relevant schemes.
However, some points should be clarified/justified:
- Why a randomized designed for an early phase with small number of patients per-group? that should be discussed (i.e. tone down)
- What is the exact performance of “PET-PDL1”??? as standard PET FDG will not give to my opinion enough strong data for a such complicated protocol (reference on sensitivity and specificity)
- If the protocol plans to include patients without huge vascular involvement, it implies that it will comprise rather small primary tumor with early metastasis? This is a limitation of this protocol (recruitment bias)
- I did see the timetable of the protocol and follow-up of patients, it seems to take a long time… i.e. heavy protocol for patients (sometimes WHO 2) after Folfirinox treatment for whom it is propose the sequence “arteriography, nivolumab IV, intratumor electroporation….etc ”
- As secondary objective there is OS and PFS: this is quite surprising for a Phase 1: PFS for a second line metastatic state over 6 patients, it seems to be a great challenge!
- What is the timing of biopsies of primary as well as metastasis?
- A separate paragraph concentrating all preclinical data would be welcome for the reader
Author Response
Point 1: Why a randomized designed for an early phase with small number of patients per-group? that should be discussed (i.e. tone down)
Response 1: We want to thank reviewer 2 for his/her thorough review and valuable comments. PANFIRE-III is first and foremost a phase-1 safety trial but the potential value of the generated immunological data is very carefully considered. In depth immunological understanding of the added value of IRE ablation to nivolumab in mPDAC, by analysis of the blood and tissue samples, will be optimally achieved when properly compared to a nivolumab control group. By randomizing between these groups, arm A (nivolumab control) and arm B (IRE + nivolumab), we aim to optimally exclude selection bias that could influence the immunological observations and potentially complicate interpretation of the results. The randomized design was not chosen to make any efficacy claims but will definitely help us to optimally identify differences in immune responses. We agree that the fact that PANFIRE-III is a randomized trial could raise expectations about potential efficacy statements. However, this is not the aim of this phase-1 trial. We have added an explanation about the evaluation of the secondary objectives related to survival on line 352-356: Furthermore, quality of life and pain questionnaires, overall survival (OS), and progression-free survival (PFS) will be recorded. Inherent to the phase-I trial design and small study sample, OS and PFS are specifically evaluated subordinate to, and in relation to the previously noted outcome measures.
We explained our choice for the randomized design on line 363-365: “Randomized inclusion will start in arm A and B (phases 1 and 2) at the same time to minimize selection bias and secure optimal qualitative immunological analyses of the added value of IRE to nivolumab in mPDAC.”
Point 2: What is the exact performance of “PET-PDL1”??? as standard PET FDG will not give to my opinion enough strong data for a such complicated protocol (reference on sensitivity and specificity)
Response 2: The 18F-BMS-986192 PD-L1 PET scan has previously been implemented in studies at our institution (Amsterdam UMC - VUmc). In the first-in-human proof-of-principle study, it was shown that in vivo imaging of PD-L1 is feasible and safe in patients with advanced stage non-small cell lung cancer (NSCLC), prior to treatment with anti-PD1 (see ref 55, Niemeijer et al) . In that study, the PD-L1 PET scans were correlated with PD-L1 expression in tumor biopsies stained by immunohistochemistry. Tumors expressing at least 50% PD-L1 had a higher uptake of 18F-BMS-986192 than lesions expressing less than 50% PD-L1. In addition, patients responding to anti-PD1 treatment had a higher 18F-BMS-986192 tracer uptake than patients without a response. Although PD-L1 expression (stained by immunohistochemistry) in tumor biopsies has previously been correlated with response and (progression-free) survival following anti-PD-(L)1 treatment, these results suggest that a non-invasive PET scan may now also be used to determine response in NSCLC patients. In addition to its non-invasive nature, PET scans allow for PD-L1 status assessment of all potential lesions instead of the one lesion that was biopsied. For pancreatic cancer, however, no clinical studies incorporating a PD-L1 PET scan are available, and hence this PET scan study is primarily used for 18F-BMS-986192 tracer validation specifically for this disease. It will provide qualitative and quantitative information before and during treatment about the tracer’s uptake in tumor and lymphoid organs, such as lymph nodes and spleen, and may give new insights in innate and adaptive immune dynamics. Indeed, type-1 IFN release by pDC upon CpG binding is expected to induce PD-L1 expression on both tumor and immune cells; additionally, T cells recruited to, and activated in, the tumor microenvironment may induce PD-L1 expression through IFN. We have elaborated on this in the discussion. See line 640-653: Imaging with 18F-BMS-986192 was previously studied in non-small cell lung cancer patients prior to treatment with anti-PD-1 mAbs and proven feasible and safe [55]. It was found that patients responding to treatment with anti-PD-1 mAbs experienced higher 18F-BMS-986192 uptake than non-responders. No clinical PET studies with PD-L1 tracers have been performed in pancreatic cancer patients yet. Hence, the incorporated explorative 18F-BMS-986192 tracer study is primarily used for tracer validation in metastatic pancreatic cancer patients as it will generate qualitative and quantitative information about the tracer’s uptake in tumor and lymphoid organs. Additionally, it might provide new insights in innate and adaptive immune dynamics following each treatment combination. Indeed, type-1 IFN release by pDC upon CpG binding may induce PD-L1 expression on both tumor and immune cells; additionally, T cells recruited to, and activated in, the tumor microenvironment may induce PD-L1 expression through IFN. It should be noted that the amount of delivered 18F-BMS-986192 is in the nanomolar quantity and far below the dose required for pharmacological effects.
Furthermore, we added a rationale for use of FGD PET with sensitivity and specificity references from line 606-631: Performance of 18F-FDG PET for diagnosis and follow-up of PDAC remains a topic of discussion and its use in the follow-up of treatment with immunotherapeutics is entirely unexplored. However, a recent meta-analysis of the use of 18F-FDG PET in combination with contrast-enhanced CT for detection of recurrent disease reported a reasonable diagnostic accuracy with pooled estimates for sensitivity and specificity of 95% and 81% respectively. Additionally, preclinical evidence supports the potential of 18F-FDG PET to monitor checkpoint inhibitor associated metabolic changes in lymphoid organs.
Point 3: If the protocol plans to include patients without huge vascular involvement, it implies that it will comprise rather small primary tumor with early metastasis? This is a limitation of this protocol (recruitment bias)
Response 3: We do not plan to exclude patients with huge vascular involvement. Irreversible electroporation is actually proposed as particularly suitable for tumors with vascular involvement as it creates nanopores in tumor cellular membranes but generally leaves collagen, glycoproteins, and endothelial cells intact. The exclusion of patients with portal vein or VMS stenosis > 70%, or any arterial stenosis (AMS, celiac artery, common hepatic artery) > 70% is specifically added to ensure that patients entering the study underwent effective stenting of one of these vessels. We added this to table 1 for completeness.
Point 4: I did see the timetable of the protocol and follow-up of patients, it seems to take a long time… i.e. heavy protocol for patients (sometimes WHO 2) after Folfirinox treatment for whom it is propose the sequence “arteriography, nivolumab IV, intratumor electroporation….etc ”
Response 4: We share the concerns expressed by reviewer 2 given the vulnerability of mPDAC patients and the frequency of study visits in the first 3 months after the first treatment. However, after these 3 months patients will shift to a less heavy schedule with only a monthly visit for nivolumab administration and a 3-monhtly ceCT scan (which is standard of care for pancreatic cancer patients in the Netherlands). Since all included patients must have at least stable disease after 8 cycles of FOLFIRINOX (effectively serving as a selection tool for relatively vital patients) we have confidence that most included patients will be fit enough for the study. Furthermore, we introduced a mandatory 4-week recuperation period after the final FOLFIRINOX cycle to ensure enough time for recovery before study start (line 281).
Point 5: As secondary objective there is OS and PFS: this is quite surprising for a Phase 1: PFS for a second line metastatic state over 6 patients, it seems to be a great challenge!
Response 5: We agree with reviewer 2 that the secondary objectives OS and PFS in this phase-1 trial will not provide us with quality survival evidence nor will OS or PFS results from this study design allow any efficacy statements for one of the treatments. Nonetheless, noting the exploratory nature of the PANFIRE-III trial, we argue that not measuring these outcomes would preclude their assessment in relation to the flowcytometric, histopathologic, proteomic and imaging assessments. We feel that OS and PFS should be included as secondary objectives but agree that the outcomes will be subordinate, and will be reported in relation to the biochemical responses, immune responses, and radiological responses; these outcomes might guide us in the design of future research. To emphasize this, we have shifted OS and PFS to the end of the paragraph and added an explanation from line 355to 3567 “Inherent to the phase-I trial design and small study sample, OS and PFS are particularly evaluated subordinate to, and in relation to the previously noted outcome measures.”
Point 6: What is the timing of biopsies of primary as well as metastasis?
Response 6: Tissue from the primary and metastatic tumor, as well as venous blood, are taken in the same session. In other words, during every biopsy session both the primary and metastatic tumor are biopsied by CT-guidance. The timing of the biopsy sessions per treatment arm is illustrated in figure 3 and corresponding legend. To clear up the timing we have added this specifically in line 381: Systemic and local immune responses are assessed by sampling venous blood and primary and metastatic tumor tissue respectively, in the same session.
Point 7: A separate paragraph concentrating all preclinical data would be welcome for the reader
Response 7: Thank you for this suggestion we have added a paragraph to distinguish preclinical from clinical data in the introduction on line 101 and discussion on line 500.

Reviewer 3 Report
This manuscript described a phase-I trial protocol for the investigator-initiated PANFIRE-III trial. The 3-arm trial aims to test the efficacy of combination therapy for metastatic pancreatic ductal adenocarcinoma, combining a local ablative therapy, irreversible electroporation (IRE), and two systematic immunotherapy drugs, CpG IMO-2125 (toll-like receptor 9 ligand) and nivolumab. And for the currently reported phase-I, the safety of the combined therapy was the primary endpoint.
Nivolumab is a fully human monoclonal immunoglobulin G4 antibody to PD-1 and used as a standard anti-PD1 drug for metastatic PDAC. The current trial applies on mPDAC patients with stable disease following at least 8 cycles of FOLFIRINOX chemotherapy. The combination therapy being tested in PANFIRE-III trial hypothesizes that combining IRE, which reduces immune suppression and stimulates a tumor-specific immune response, with PD-1 checkpoint inhibition using nivolumab, preceded by effective DC priming through intra-tumoral injection of IMO-2125 might establish in vivo immunization and durable treatment results in mPDAC patients.
The manuscript is clearly written for the most part, and contains all relevant elements for a protocol report. The mechanism of each therapy is introduced with a right amount of detail, with some comparative results from citations to motivate why a specific technique or drug is chosen. The study design was also well described. I believe it is an interesting protocol to publish in this journal but just have a few questions/comments as below.
The time sequences are slightly different for Arm C compared with for Arms A and B. While I understand that this is due to the additional CpG, the non-uniform lab and imaging collecting timepoints may compromise the trial. Especially for the efficacy testing in the 2ndary endpoint of the current trial and in the later phases, this will be an issue.
For ceCT, how will the scanning range be set? In other words, there could be new mets out of the scanning range if it is limited around the abdomen. Will new uptake regions in PET be used to expand the ceCT scanning range?
The writing is good but there are a few typos/errors in the text that need to be proofread and corrected.
Author Response
Point 1: The time sequences are slightly different for Arm C compared with for Arms A and B. While I understand that this is due to the additional CpG, the non-uniform lab and imaging collecting timepoints may compromise the trial. Especially for the efficacy testing in the 2ndary endpoint of the current trial and in the later phases, this will be an issue.
Response 1: We want to thank reviewer 3 for his/her kind introduction and thorough review of our manuscript. We understand the concern and have spent quite some time discussing this internally as well. The reason we eventually chose to defer the study assessments with 1 week in arm C is because the PANFIRE-III study is primarily designed to investigate the added value of IRE to immunotherapy. We therefore centered the study assessments around the IRE ablation rather than around the first study treatment. We performed an immune monitoring study in locally advanced pancreatic cancer patients that showed a peak IRE immune stimulatory effect 14 days after treatment. Consequently, we want to asses IRE in combination with Nivolumab (arm B) and compare this with combination of IRE with CpG (arm C) on this peak moment. Should we have assessed all study arms at the same time points we would have introduced bias and could possibly miss this IRE peak effect. The same rationale holds for assessment of the adaptive immune response at 5 (arm A and B) and 6 (arm C) weeks after treatment start. To evaluate the added value of IRE to Nivolumab and to CpG + Nivolumab the time frame between each IRE treatment and study assessment should be the same: 5 weeks. For the later study phases, after 3 months, all study assessments (ceCT and questionnaires every 3 months), will be performed on the same time points for all study arms.
We have tried to explain our rationale for the differential time sequences in the discussion on line 537-545:
“Assessments of the post treatment immune response are specifically timed for optimal comparison between the treatment arms. Blood and tissue are sampled 2 (arm A and B) or 3 weeks (arm C), and 5 (arm A and B) or 6 weeks (arm C) after start of the treatment for assessment of the innate immune response to each individual treatment (arm A: nivolumab, arm B: IRE, arm C: CpG + IRE), and the adaptive immune response to each individual treatment in combination with Nivolumab, respectively. To optimally compare the immune effects induced by IRE to the addition of CpG and/or Nivolumab, identical timing between IRE ablation, and blood and tissue sampling in arm B and C was given precedence. Hence, sampling in arm C is deferred 1 week.”
Point 2: For ceCT, how will the scanning range be set? In other words, there could be new mets out of the scanning range if it is limited around the abdomen. Will new uptake regions in PET be used to expand the ceCT scanning range?
Response 2: Thank you for this question; we indeed agree that restricting the scanning range to the abdominal area means we will surely miss metastases of the thoracic cavity, mainly those in the lungs. Hence, the FDG and PD-L1 PET-CTs will be whole body scans and the scanning range of ceCT scans will be set to cover the abdomen and thorax. This was added on line 447. The final PET scan, according to the protocol, will be made at 3 months post study start. Thereafter, radiological follow-up consists of ceCT scans every 3 months (T=6 months, 9 months, 12 months etc.). We acknowledge that the brain area is not included in these ceCT scans, but with brain metastasis occurring in only approximately 0.5% of patients, this is not standard clinical practice in the Netherlands.
Point 3: The writing is good but there are a few typos/errors in the text that need to be proofread and corrected.
Response 3: Thank you for your alertness in this matter. We have applied an extra review round to find and solve these errors.

Round 2
Reviewer 1 Report
thank you for improving the manuscript